# Ibuprofen: Toxicology and Biodegradation of an Emerging Contaminant

**DOI:** 10.3390/molecules28052097

**Published:** 2023-02-23

**Authors:** Janet Jan-Roblero, Juan A. Cruz-Maya

**Affiliations:** 1Laboratorio de Biotecnología Ambiental, Departamento de Microbiología, Escuela Nacional de Ciencias Biológicas, Instituto Politécnico Nacional, Mexico City 11350, Mexico; 2Unidad Profesional Interdisciplinaria en Ingeniería y Tecnologías Avanzadas, Instituto Politécnico Nacional, Mexico City 07340, Mexico

**Keywords:** emerging contaminant, ibuprofen, toxic compound, biodegradation, bacteria

## Abstract

The anti-inflammatory drug ibuprofen is considered to be an emerging contaminant because of its presence in different environments (from water bodies to soils) at concentrations with adverse effects on aquatic organisms due to cytotoxic and genotoxic damage, high oxidative cell stress, and detrimental effects on growth, reproduction, and behavior. Because of its high human consumption rate and low environmental degradation rate, ibuprofen represents an emerging environmental problem. Ibuprofen enters the environment from different sources and accumulates in natural environmental matrices. The problem of drugs, particularly ibuprofen, as contaminants is complicated because few strategies consider them or apply successful technologies to remove them in a controlled and efficient manner. In several countries, ibuprofen’s entry into the environment is an unattended contamination problem. It is a concern for our environmental health system that requires more attention. Due to its physicochemical characteristics, ibuprofen degradation is difficult in the environment or by microorganisms. There are experimental studies that are currently focused on the problem of drugs as potential environmental contaminants. However, these studies are insufficient to address this ecological issue worldwide. This review focuses on deepening and updating the information concerning ibuprofen as a potential emerging environmental contaminant and the potential for using bacteria for its biodegradation as an alternative technology.

## 1. Introduction

Diseases have been a constant health problem for humanity, and pharmaceutical products are a means to control them. An inflammatory response often accompanies diseases in humans. Therefore, drugs with anti-inflammatory activity are highly frequently used, creating a constant demand worldwide and increasing their production. The medical control of anti-inflammatories is generally unregulated, with some controlled by prescription and others not. Among pharmaceutical compounds not requiring a medical prescription is the group of nonsteroidal anti-inflammatory drugs (NSAIDs), including paracetamol, naproxen, and ibuprofen. Ibuprofen is a globally recognized anti-inflammatory with high human consumption rates, since it is frequently prescribed as an antipyretic, analgesic, and anti-inflammatory to reduce fever, headache, muscle pain, menstruation discomfort, neurological pain, and post-surgical pain. Ibuprofen inhibits cyclooxygenase, which converts arachidonic acid into cyclic endoperoxides that are transformed into prostaglandins and thromboxanes, which are inflammation mediators. Inhibiting cyclooxygenase and subsequent prostaglandin synthesis reduces the release of inflammatory substances and mediators, preventing the activation of pain receptors (nociceptors). Ibuprofen is insoluble in water and has the chemical structure 2-(4-isobutylphenyl) propanoic acid, containing an aromatic ring with isobutyl substitutions and propanoic acid, and two enantiomeric forms (R and S).

High ibuprofen consumption has been considered a potential environmental risk since annual NSAID production oscillates in a few kilotons [1]. This drug is distributed across societies that use it for specific medical purposes. It is metabolized within the body to generate ibuprofen derivatives. In the USA, UK, and Poland, ibuprofen consumption is approximately 300, 162, and 58 tons per year, respectively [2]. There has been a recent increase in ibuprofen consumption in many Nordic countries. However, its consumption remains low in Norway and Denmark, at approximately 2.0 and 1.15 tons per year, respectively [3].

Different sources introduce metabolized ibuprofen into the environment, where it accumulates in natural environmental matrices. Metabolized ibuprofen is mainly discharged into the water systems of different communities, accumulating in the aquatic environment or soil. However, non-consumed expired drugs are often dumped in domestic (household) garbage cans, ultimately ending up in municipal garbage cans where they are a latent potential soil contaminant. Ibuprofen can also be used in veterinary medicine. After consumption, it is released on the ground in the form of animal urine or feces.

The problem of drugs as contaminants, particularly regarding ibuprofen, is complicated because few strategies or programs confine or extract pharmaceutical compounds or apply successful technologies for their removal in a controlled and efficient manner. Drugs present in municipal wastewater are not removed by the strategies used in wastewater treatment plants. Furthermore, the problem is exacerbated by the great variety of drugs with different chemical structures and physicochemical characteristics that make it difficult to develop specific technologies for each. In addition, some drugs are difficult to degrade in the environment or via microorganisms due to their physicochemical properties, including ibuprofen.

Currently, some experimental studies focus on the problem of drugs as potential environmental contaminants. However, they are insufficient to address this ecological problem worldwide, since ibuprofen’s introduction into aquatic or soil systems is monitored in different countries. This review focuses on deepening and updating the information on ibuprofen as a potential emerging environmental contaminant and the potential use of bacteria for its biodegradation as an alternative technology.

## 2. Presence of Ibuprofen in the Environment (Emerging Pollutant)

As mentioned above, the high consumption of ibuprofen and other pharmaceuticals has caused them to become of environmental concern. Consumed or expired ibuprofen is discarded by different routes into ecosystems, where its presence can have adverse environmental effects. Until now, ibuprofen has been considered an emerging contaminant due to its constant presence in different environments (from water bodies to soils) and its adverse effects on the health of animals, mainly aquatic ones. In addition, since none of the applied technologies in conventional wastewater treatment plants completely remove ibuprofen, it will be present in their outlet water [1].

Ibuprofen can enter ecosystems through different routes, of which wastewater effluents are the main one [4], including aquifer effluents from veterinary facilities, domestic premises, hospitals, and drug-production factories (Figure 1) [5,6]. Ibuprofen has been detected in sediment and surfaces, ground soil, and agricultural fields [1]. One of the main ways in which ibuprofen enters the soil is its use in veterinary medicine, where animals administered ibuprofen discard its physiological products into the soil. This organic waste becomes raw material for sewage sludge or manure fertilizers, which can leach or spread in fields. Therefore, ibuprofen in the soil can be mobilized and find its way into the groundwater (Figure 1) [4]. Finally, acquiring ibuprofen without a medical prescription is common, and its by-products reach municipal wastewater through the urine or feces of consumers. In other cases, expired and unconsumed ibuprofen is sent to municipal landfills, entering the ground as a contaminant [2] (Figure 1).

Ibuprofen can be introduced into the environment in its original or modified form by the organism that consumed it (Figure 1). The ibuprofen that reaches the environment in its original form is mainly from unconsumed or expired pills in human waste in municipal landfills. Its modified forms (transformation products) reach the environment through excretion in urine or feces after consumption by humans or animals [2]. Humans metabolize ibuprofen in two ways: by its binding to glutathione, glucuronic acid, or sulfates (conjugated), or by its hydroxylation [2]. Orally administered ibuprofen is rapidly distributed throughout the body in humans, with most (99%) bound to plasma albumin. Fifteen percent of orally administered ibuprofen is excreted in its original, conjugated, or hydroxylated form (2-, 3-, and 1-hydroxyibuprofen) or as carboxyibuprofen or carboxyhydratropic acid [2]. In the environment, glucuronide-conjugated ibuprofen can be hydrolyzed, releasing free ibuprofen [2,7]. However, ibuprofen in wastewater is transformed into more toxic intermediate compounds during the chlorination process in treatment plants [8], including 1-(4-isobutylphenyl)-1-ethanol, 4-isobutyricbenzaldehyde [9], 2-[4-(1-hydroxyisobutyl)phenyl] propionic acid, 1-ethyl-4-(1-hydroxy) isobutylbenzene, 4-ethylbenzaldehyde, and 4-ethylphenol [10].

Little interest exists in establishing drug detection programs in ecosystems. The available data on ibuprofen in ecosystems were generated by research studies. Some countries have reported the presence of various drugs in water sources, including Greece and India, where 11 drugs, including ibuprofen, were detected in hospital and municipal wastewater treatment plants [11,12]. In Pakistan, ibuprofen has been reported in zones or areas close to pharmaceutical factories [13]. Therefore, the entry of ibuprofen into the ecosystems is an unattended contamination problem that represents a concern for our environmental health system and requires further attention.

To date, there is no parameter or limit set for ibuprofen in the environment. Reports indicate that its concentration varies in different countries and depends on the environmental sample. In the case of aquatic environmental samples, the concentrations are generally at μg/L. For example, in wastewater treatment plants, ibuprofen has been detected at concentrations of 0.004–603 μg/L in Bosnia, China, Croatia, Greece, Herzegovina, Korea, Serbia, Sweden, Switzerland, and the UK [14]. In wastewater samples, concentrations of 45, 1.38, 5.78, or 703–1673 μg/L have been reported in Canada, South Africa, Belgium, and Pakistan, respectively [15,16]. In surface waters, average concentrations were 0.98, 8.0, 1417, 1.0–67, 15–414, and 5.0–280 μg/L for Canada, France, China, Greece, Korea, and Taiwan, respectively [14,17]. In environmental sludge samples, concentrations differed significantly between South Africa (0.009 μg/kg) and Pakistan (2053–6064 μg/kg) [18], indicating that ibuprofen concentration in this ecosystem depends on the management or control each country has over the drug.

Soil samples had widely varying ibuprofen concentrations ranging from 321–610 μg/kg, and in agricultural soils irrigated with ibuprofen-contaminated wastewater, the concentration was of 0.213 μg/L [13,19]. A similarly wide range of ibuprofen concentrations has been detected in groundwater, ranging between 3 ng/L and 395 ng/L on the European continent [14]. Ibuprofen transformation products have also been detected in influent wastewater treatment plants at concentrations of 20.24 ± 7.1 ng/L for carboxyibuprofen, 1091 ± 814 ng/L for 1-hydroxyibuprofen, and 7768 ± 2693 ng/L for 2-hydroxyibuprofen [20]. In Spain, in a wastewater treatment plant located in Girona, which collects urban, domestic and industrial wastewaters, the maximum concentrations in influent wastewater samples were 13.74, 5.8, 38.4, and 94.0 μg/L for ibuprofen, 1-hydroxyibuprofen, carboxyibuprofen and 2-hydroxyibuprofen, respectively; whereas maximum levels in effluent wastewater samples were 1.9, 1.4, 10.7, and 5.9 μg/L for ibuprofen, 1-hydroxyibuprofen, carboxyibuprofen and 2-hydroxyibuprofen, respectively. These compounds were also detected at high levels in the Ter River, specifically carboxyibuprofen, which was detected up to 3.9 μg/L [21].

Some general technologies are applied in some wastewater treatment plants to eliminate emerging contaminants, including pharmaceuticals. These technologies are based on filtration (biofiltration or nanofiltration), adsorption (activated carbon nanocomposites), reverse osmosis, and oxidation [14,22]. In a hospital wastewater treatment plant in Germany, nanofiltration and reverse osmosis were applied at pilot level, the nanofilter membrane was 300–400 Da, the reverse osmosis membrane was 100–150 Da, and the effluent feed was 20 and 36 L per 2 h; under these conditions, ibuprofen was eliminated at 85.7% and 95.8% by nanofiltration and reverse osmosis, respectively [23]. In the case of adsorption, using chemically surface-modified activated carbon cloth, at pH 3, 25 °C, 10 h, 250 rpm, the ibuprofen adsorption capacity was 491.9 mg/g [24]. The polyaniline nanocomposite (PANI 900), a catalytic ozonation method classified as an oxidation process, eliminated significantly more ibuprofen (95%) at pH 10 for 20 min than the original process [25]. Finally, the application of photocatalysis coupled to UV-Vis-NIR irradiation has potential for the degradation of antibiotics and possibly for other pharmaceuticals [26]. Therefore, these technologies are promising and can provide adequate results for eliminating pharmaceuticals. However, it is important to consider that various pharmaceuticals with different physicochemical properties are mixed in wastewater, which can complicate their efficient disposal [27]. Consequently, several technologies will need to be consecutively applied to achieve greater efficiency in eliminating pharmaceuticals. Few wastewater treatment plants currently apply these new technologies. Therefore, since these centers are the first barrier preventing the entry of many contaminants into the environment, drugs are continually being introduced into the different ecosystems, accumulating for long periods and causing damage to aquatic organisms [28]. The problem is complicated because some drugs are recalcitrant due to their low water solubility and resistance to biodegradation, favoring their persistence [29].

## 3. Ibuprofen Toxicity in Organisms

A few studies have examined the toxicity of ibuprofen in living beings [30], focusing mainly on its toxigenic capacity in living organisms in aquatic ecosystems. Ibuprofen’s characteristics (high lipophilic degree and low biodegradation) favor its bioaccumulation in the environment [31], and its biological activity suggests that it is harmful to several aquatic species. The environmental accumulation of pharmaceutical compounds with a biological action such as ibuprofen’s could increase their toxicity in aquatic organisms [32].

Ibuprofen toxicity studies have been performed in organisms from the highest to the lowest biological hierarchical levels (sub-organisms), finding that it caused adverse effects. The ibuprofen concentration exposed to the organism will generate damage. An acute effect is produced in organisms by exposure to high ibuprofen concentrations (>100 mg/L). It can produce sub-lethal effects at concentrations between 10 and 100 mg/L. The average ibuprofen concentration in ecosystems is between 0.2 and 8.0 μg/L. Prolonged ecological exposure to ibuprofen in aquatic animals can have chronic and even drastic effects, including cytotoxic and genotoxic effects, high oxidative cell stress, and adverse effects on growth rate, reproduction, and behavior. These effects have been observed in vivo and in vitro studies simulating an ecosystem contaminated with ibuprofen [30]. It has also been suggested that ibuprofen in living organisms can be biotransformed into intermediate compounds with more toxic characteristics than ibuprofen itself [1]. For example, ibuprofen conjugated with diacylglycerol inhibited cell division and the nondisjunction of some chromosome pairs [33]. In experiments carried out with batch activated sludge, it was shown that ibuprofen was degraded and 1,2-dihydroxyibuprofen was detected, in addition to 2-hydroxyibuprofen and 1-hydroxyibuprofen remaining in solution, whereas carboxyibuprofen disappeared faster than hydroxylated transformation products, which had the lowest removal rates [21].

Biological models used to study the damaging effect of drugs, including antibiotics, include zebra mussels (*Dreissena polymorpha*) and zebrafish (*Danio rerio*). In zebra mussels, ibuprofen for seven days had lipid peroxidation and disrupted enzymatic activities [34]. Prolonged exposure to ibuprofen caused a chronic effect, provoking genetic and cellular damage [1]. Another effect was cellular oxidative stress, since its exposure imbalanced the activities of catalase (CAT), superoxide dismutase (SOD), glutathione peroxidase, and the phase II detoxifying enzyme glutathione S- transferase (GST) [1]. In the zebrafish, ibuprofen administration at environmental concentrations (0.1–11 μg/L) altered the antioxidant defense system and liver protein carbonylation, and increased blood vitellogenin-like protein levels and lactate dehydrogenase activity [35]. It has been observed that ibuprofen is a potential generator of radical species and therefore increases oxidative stress; this effect has been observed in all evaluated tissues of zebrafish after long-term exposure to environmentally relevant concentrations [36]. Hatching alterations, head malformation, skeletal deformities, hypopigmentation, pericardial edema, and heart rate impairment were also present in the zebrafish embryos, in addition to oxidative stress [37]. In the larval stage of zebrafish at 5 days post fertilization, ibuprofen reveled perturbations of thermal preference behavior upon exposure during embryogenesis [38]. These results show that ibuprofen, even at low concentrations similar to those detected in the environment, induced changes in zebra mussels and zebrafish models.

A few studies have focused on ibuprofen’s toxic effect on freshwater fish species. One study found that ibuprofen exposure caused changes in the kidneys of freshwater *Oncorhynchus mykiss*, including hyalinosis, increased oxidative stress, and changes in heat shock protein 70 expression. In addition, it affected their liver, causing dystrophy, cellular congestion, and inflammation [39]. In bivalve mollusks (*Unio tumidus*; often used as bioindicators of freshwater quality), exposure to a sub-chronic environmental ibuprofen concentration (at 0.8 µg/L) for 14 days decreased the protein carbonyl concentration, induced cholinesterase, and reduced lysosomal integrity in their digestive glands [40]. Chronic ibuprofen toxicity has been shown in other biological freshwater species models (*Oryzias latipes, Daphnia magna, and Moina macrocopa*), with changes in hormonal balance observed under in vitro conditions, including increased 17β-estradiol production and aromatase activity, but decreased testosterone production [41]. In *Oryzias latipes*, exposure to ibuprofen at 0.1 μg/L impaired reproduction, induced vitellogenin in males, decreased the number of broods per pair, and increased the number of eggs per brood [41]. In *D. magna*, exposure to ibuprofen concentrations similar to those detected in ecosystems significantly decreased the total number of eggs and broods per female and body length [42]. Furthermore, after ibuprofen exposure, *D. magna* showed increased oxidative stress through CAT, SOD, and GST activities. However, exposure to ibuprofen at concentrations of 1–100 ng/L decreased the behavioral activity of *Gammarus pulex* [43]. Ibuprofen administration at 1 μg/L, 2 μg/L and 4 μg/L during the whole-life-cycle of *Daphnia magna* showed slow growth, late maturation and longer life, with a higher reproduction rate and smaller broods, but larger neonates [44].

Ibuprofen alters blood components, which has been proposed as an initial toxicity parameter. Indeed, the Indian major carp (*Cirrhinus mrigala*), an aquatic organism, showed increased hemoglobin, hematocrit, mean cellular volume, mean cellular hemoglobin, leukocytes, plasma glucose, and alanine transaminase after ibuprofen administration at 14.2 mg/L for 24 h [45]. The same changes in blood were reported in *D. polymorpha* [30] and *Ruditapes philippinarum* [46]. In addition, ibuprofen affected fertility, with a significant reduction in fertilization observed in *Psammechinus miliaris* after ibuprofen treatment [47]. These results show that ibuprofen can have different toxic effects on aquatic organisms, suggesting that this may also occur in other aquatic and non-aquatic species.

Ibuprofen toxicity extends to other aquatic animals which are not models of drug toxicity; such as in the African catfish *Clarias gariepinus,* which after exposure to environmental ibuprofen concentration was observed to have histopathological deformities in the gills, liver, and kidney, with alterations such as severe secondary lamella necrosis (SLN), epithelial lifting (EL), mild deformity of the secondary lamella (DSL), mild secondary lamella necrosis (MLN), and mild vascular congestion in the liver and kidney [48]. SOD activity decreased in the crustacean *Gammarus pulex* after exposure to ibuprofen, as well as CAT activity at 24 and 96 h, and at 96 h there was also an increase in acetylcholine esterase AChE activity [49]. In the clam *Scrobicularia plana*, ibuprofen exposure for 21 days produced biochemical alterations (SOD, CAT, lipid peroxidation (LPO) and AChE) in gills and digestive glands, indicating oxidative stress and damage to lipids [50]. In adult females of the estuarine crab *Neohelice granulate*, ibuprofen exposure (10 mg/L) for 60 days caused a very low content of total vitellogenic proteins in the ovary, and at 90 days, a decrease in the proportion of vitellogenic oocytes was observed [51]. The diatom *Phaeodactylum tricornutum* exposed to high concentrations of ibuprofen (0.1 and 0.3 mg/L) affected its growth and inhibited the electron transport chain, producing less energy [52].

In other events, ibuprofen at ambient concentrations negatively impacted tadpoles’ growth and development by significantly delaying their time to metamorphosis and reduced body weight, and a mutagenic analysis of blood revealed a significant increase in the frequency of cellular and nuclear abnormalities of *Bufo bufo* tadpoles [53]. Ibuprofen toxicity also extends to plants; in cowpea (*Vigna unguiculata*), ibuprofen caused a reduction in the following parameters: shoot and root lengths, fresh and dry weights, leaf area, and chlorophyll a and b, carotenoid, total chlorophyll, mineral (K and Mg), glutathione reductase, and soluble protein contents, and, simultaneously, increases in Ca and Mn contents, sodium translocation from roots to shoots, H_2_O_2_, malondialdehyde, superoxide dismutase, catalase, and ascorbate peroxidase [54]. The allerquin fly *Chironomus riparius* exposed to ibuprofen developed a substantial effect on survival, as well as alterations in the mRNA levels of EcR, Dronc, and Met (endocrine system), hsp70, hsp24, and hsp27 (stress response), and Proph and Def (immune system), and finally an increase in the activity of GST and LPO [55].

Ibuprofen toxicity in bacteria has been reported. In freshwater, ibuprofen exposure changed the intestinal microbiota of the fish *Oncorhynchus mykiss*, where Gram-positive bacterial species showed a general overgrowth, and the Fusobacteria/Firmicutes bacterial ratio was significantly altered [39]. In addition, altered bioluminescence was observed in the *Allivibrio fischeri* bacterium at ibuprofen concentrations of 19.1 and 37.5 mg/L for 15 min [56]. In an activated sludge system, ibuprofen significantly reduced the microbial diversity and changed the bacterial community structure; for example, some denitrifiers (*Denitratisoma* and *Hyphomicrobium*) increased significantly, while *Nitrospira* significantly decreased under ibuprofen stress; in addition, different phylogenetic populations had different responses to ibuprofen, such as an increase in Chloroflexi and a decrease in Proteobacteria [57].

On the other hand, the structure of ibuprofen confers to its S-(+)- and R-(−)-ibuprofen enantiomers different activity and toxicity, such as R-(−)-ibuprofen, which stereoselectivity inhibits hepatic mitochondrial beta-oxidation in rats; however, mitochondrial respiration is moderately inhibited by S-(+)- and R-(−)-ibuprofen [58]. The S-(+)-ibuprofen enantiomer is not toxic to chondrocytes or synoviocytes [59]. Studies suggest that S-(+)-ibuprofen has greater anti-inflammatory activity than R-(−)-ibuprofen, lower toxicity, greater clinical efficacy, and less variability in therapeutic effects. In addition, S-(+)-ibuprofen reduces the development of cancer and prevents the development of neurodegenerative diseases [60]. Regarding the environment, the racemic mixture of ibuprofen has been detected in the Guadalquivir River basin (South Spain) [61]. In aquatic water-sediment systems, the racemic mixture of ibuprofen has been detected, and it has been observed that the R-(−)-enantiomer has a better degradation than the S-(+)-enantiomer. It has also been observed that the half-life of ibuprofen increased from 5.8 to 10.1 days in the sediment [62]. By studying the effect of ibuprofen enantiomers on aquatic animals, they show that the liver of rainbow *Oncorhynchus mykiss* processes these enantiomers more slowly [63]. Adult zebrafish (*Danio rerio*) brain after exposure to R-(−)-/S-(+)-/rac-IBU at 5 μg/L for 28 days showed stereoselective alterations in the 45 identified biomarkers from the untargeted metabolomics analysis, with 22 amino acids quantitated, and 3 antioxidant enzymes [64], indicating that ibuprofen enantiomers have different effects. Assays with the algae *Chlorella pyrenoidosa* showed that S-(+)-ibuprofen is more toxic than R-(−)-ibuprofen, but R-(+)-ibuprofen bioaccumulated and degraded from this organism, and no interconversion occurs between the two enantiomers [65]. S-(+)-ibuprofen and the racemic mixture inhibited the growth of the green alga *Scenedesmus obliquus* as well as the cellular ultrastructure, causing the plasmolysis, deformation and disintegration of chloroplasts and low chlorophyll and carotenoid content [66]. A toxic effect on bacteria has also been reported with both enantiomers [67]. This demonstrates that there is a stereoselectivity of ibuprofen on toxicity in organisms.

Ibuprofen toxicity is widespread in both higher and sub-organisms, indicating the importance of monitoring ibuprofen-contaminated environments to accurately assess its toxicological effects on communities of organisms. Such data are necessary for proposing regulatory strategies or technologies to eliminate ibuprofen and reduce its environmental impact.

## 4. Ibuprofen Biotransformation/Biodegradation (Biodegradation Pathways)

An environmentally friendly alternative for eliminating pharmaceuticals is biodegradation by microorganisms, such as bacteria (bioremediation). The bioremediation strategy is promising but has not yet been implemented at the wastewater treatment plant level. While ibuprofen biodegradation assays have been developed at the laboratory level, their scaling is complex because microorganisms require special conditions, such as optimum temperature and pH, an additional carbon source, and adequate xenobiotic levels to induce the appropriate enzymes for degradation; these conditions rarely occur in the environment. For example, ibuprofen removal is related to the pH of the medium since it was greater at pH 7.2 than at pH 8, at which no ibuprofen was removed [68]. Additionally, microorganisms with the potential to degrade ibuprofen often must adapt to environmental conditions. This adaptation process requires long periods that slow the contaminated ecosystem’s bioremediation [69,70]. Although it is difficult to control environmental conditions, the alternatives to achieve the adaptation of ibuprofen biodegrading microorganisms would be through the isolation of autochthonous microorganisms with the capacity to degrade ibuprofen from the contaminated region, since they would be adapted to environmental conditions. Another option is to genetically modify the autochthonous microorganisms or the application of the enzymes that participate in the degradation of ibuprofen. Finally, another impractical strategy is ex situ degradation, taking massive samples from the contaminated place and transferring them to a treatment plant where the optimal conditions for microorganisms are controlled.

Ibuprofen has phenylacetic acid (PAA) in its base structure. Ibuprofen’s chemical structure confers high resistance to biodegradation due to an aromatic ring with branched substitutions in the *para* position. Compounds with aromatic rings are usually more resistant to degradation than aliphatic compounds. Therefore, ibuprofen’s structural characteristics make it refractive to degradation by microorganisms. However, while ibuprofen’s physicochemical properties indicate that it is a highly mobile compound in aquatic environments [1], experimental data shows that its persistence is lower than other drugs, indicating that it is degraded.

Identifying metabolic pathways for ibuprofen degradation is possible using bacteria that can biodegrade ibuprofen. However, these pathways remain under study. To date, some ibuprofen degradation pathways have been identified. Hydroxylation is the primary mechanism, which is either via its binding to coenzyme A (CoA) or via the direct trihydroxylation of its aromatic ring [71,72]. As a general rule, ibuprofen biodegradation occurs via several pathways, depending on the microorganism. Different studies on ibuprofen biodegradation by bacteria have shown that it can be hydroxylated on its side chains (isobutyl and propanoic) or aromatic ring. Aliphatic monooxygenases perform the hydroxylation of the isobutyl chain to produce 2-hydroxyibuprofen or 1-hydroxyibuprofen, which are then dihydroxylated by acyl-CoA synthase to create hydroquinone derivatives. Hydroxylation of the propionic acid chain leads to CoA being added to propionic acid’s carboxylic group by acyl-CoA synthase to create ibuprofen-CoA, which is then hydroxylated by dioxygenases and deacetylated to produce p-isobutylcatechol. Another ibuprofen biotransformation route is by reducing propionic acid’s carboxyl group to produce ibuprophenol and then ibuprophenol-acetate. Finally, the hydroxylation of ibuprofen’s aromatic ring by aliphatic monooxygenases creates trihydroxyibuprofen, destabilizing it due to cleavage via the meta-cleavage pathway [71,72] (Figure 2).

Notably, the hydroxylated and carboxylated transformation products formed by the different ibuprofen biodegradation pathways are considered as being more toxic than the ibuprofen. These transformation products are detected and identified at the laboratory level in wastewater treatment plants [7,73,74]. Therefore, it is necessary to identify bacteria that can more efficiently biodegrade ibuprofen until it is mineralized.

## 5. Ibuprofen-Degrading Bacteria

Bacteria with ibuprofen biodegrading potential have been identified in recent decades. However, while laboratory-level data is available, they remain to be unused in wastewater treatment plants to eliminate ibuprofen. The first bacterium with ibuprofen biodegrading potential was the strain *Nocardia* sp. NRRL 5646, which grew using ibuprofen at an initial concentration of 1000 mg/L for 120 h as the sole carbon and energy source. The transformation products detected in the culture medium were ibuprophenol, ibuprofen acetate, and benzoic acid derivatives [75], indicating that this bacterium uses this degradation pathway (Figure 2).

The bacterial strain *Sphingomonas* sp. Ibu-2 was isolated from a water sample taken from a wastewater treatment plant, which grew for 80 h in a mineral medium supplemented with ibuprofen at 500 mg/L under aerobic conditions. This bacterium has the enzymatic machinery to remove the propionic acid chain from ibuprofen, forming catechols or methylocatechols such as isobutylcatechol; the same happens with other PAA derivatives, such as 2-phenylpropionic acid, 3- and 4-tolylacetic acids, and 2-(4-tolyl) propionic acid [71].

Biochemical and genetic studies of the *Sphingomonas* sp. Ibu-2 determined its ibuprofen degradation components. The construction of a total DNA library of *Sphingomonas* sp. Ibu-2 cloned into fosmids enabled the identification of the *ipfABDEFHI* operon composed of seven genes participating in this biodegradation pathway for ibuprofen and other PAA compounds. The functions of the operon genes have been described: the *ipfA* and *ipfB* genes encode two dioxygenase subunits that attack the aromatic ring; the *ipfD* gene encodes a sterol carrier protein X thiolase; and the *ipfF* gene encodes a CoA ligase enzyme. However, the function of the protein encoded by the *ipfE* gene remains unknown. In addition, the *ipfH* and *ipfI* genes encode a ferredoxin reductase and aromatic dioxygenase, respectively (Figure 2). With the identification of these genes, it was proposed that the ibuprofen degradation mechanism in *Sphingomonas* sp. Ibu-2 is as follows: the enzyme CoA ligase (*ipfF*) attaches CoA to ibuprofen; the multicomponent oxygenase IpfABHI then dihydroxylates ibuprofen-CoA to form 1,2-cis-diol-2-hydroibuprofen-CoA; IpfD and IpfE then degrade this molecule to form 4-isobutylcatechol and propinyl-CoA [71,72,76]. Currently, the ibuprofen biodegradation pathway is thought to have two stages. The first (upper) stage comprises the formation of 4-isobutylcatechol and propinyl-CoA [71,72,77]. The second (lower) stage involves breaking the ibuprofen ring, creating compounds that enter the tricarboxylic acid (TCA) cycle.

To identify the genes participating in the lower ibuprofen degradation pathway, those flanking the *ipfABDEFHI* operon in *Sphingomonas wittichii* MPO218 [78] and *Sphingopyxis granuli* RW412 [77] were identified and assigned as putative pathway participants. The lower ibuprofen biodegradation pathway has been partially described in the bacterium *Rhizorhabdus wittichii* MPO218, with propionyl-CoA carboxylase (encoded by the *pccAB* gene) converting propionyl-CoA to S-methylmalonyl-CoA, which then enters the TCA cycle. Isobutylcatechol from the upper ibuprofen biodegradation pathway is converted by 4-isobutylcatechol 2,3 dioxygenase (encoded by the *ipfL* gene) to 5-isobutyl-2-hydroxymuconate semialdehyde, which is then converted by 5-isobutyl-2-hydroxymuconate semialdehyde dehydrogenase (encoded by the *ipfM* gene) to 5-isobutyl-2-oxymuconate. The bacterium *Rhizorhabdus wittichii* MPO218 has other genes with undetermined functions that are thought to participate in the degradation steps: a tautomerase (*ipfP*), a decarboxylase (*ipfO*), a hydratase (*ipfN*), an aldolase (*ipfS*), an aldehyde dehydrogenase (acylating; *ipfQ*), and an acyl-CoA dehydrogenase (*ipfT*) [79]. The final products could be further metabolized by beta-oxidation [80] and the TCA cycle (Figure 2).

The *ipfABDEF* operon has been used as a genetic marker to search for other putative ibuprofen-degrading bacteria through their genome analyses. It has been found in the genomes of ten strains, of which five are from the *Cycloclasticus* genus: *C. zancles* 78-ME, C. *pugetii* PS-1, *Cycloclasticus* sp. PY97M, *Cycloclasticus* sp. DSM 27168, and *Cycloclasticus* sp. P1. While the abilities of these five strains to use ibuprofen as a carbon and energy source remains unknown, they can degrade naphthalene and phenanthrene under aerobic conditions, which are polycyclic aromatic compounds [81,82]. Other bacterial genera with the *ipfABDEF* operon in their genomes are *Pseudoxanthomonas* spadix BD-a59, *Rhodospirillales* bacterium 69-11, Comamonadaceae bacterium SCN 68-20, *Noviherbaspirillum* sp. Root189, and Gammaproteobacteria bacterium TR3.2. These bacteria were isolated from various sources. *Pseudoxanthomonas* spadix BD-a59 and Comamonadaceae bacterium SCN 68-20 were isolated from oil-contaminated environments, suggesting that they may use polycyclic aromatic compounds. *Rhodospirillales* bacterium 69-11 and *Noviherbaspirillum* sp. root189 were isolated from bioreactors. Gammaproteobacteria bacterium TR3.2 was isolated from the root of *Arabidopsis thaliana* [83,84]. In experiments, these bacteria could degrade polycyclic aromatic compounds [83,85,86]. The presence of the *ipfABDEF* operon in their genomes indicates that they could potentially degrade ibuprofen because they also degrade other cyclic compounds.

The bacterial strain *Variovorax* sp. Ibu-1 was isolated from activated sludge samples. It was found to degrade ibuprofen in a mineral salt medium at a concentration of 200 mg/L when incubated for 75 h under laboratory conditions. In addition, its ibuprofen biodegradation mechanism was determined, consisting of the trihydroxylation of ibuprofen’s aromatic ring, which is later used as a substrate to break the ring via the meta-ring fission enzyme pathway [7]. This ibuprofen trihydroxylation pathway was confirmed by the detection of ibuprofen trihydroxylation in the presence of 3-fluorocatechol, an inhibitor of the meta-ring fission enzyme pathway (Figure 2). Direct hydroxylation of ibuprofen’s ring occurs under environmental conditions since the trihydroxylated metabolite was present in sewage sludge contaminated with ibuprofen, suggesting that this catabolic pathway rather than the CoA ligation pathway is likely the most relevant under ambient conditions [7,87].

Another bacterium that could degrade ibuprofen at an initial concentration of 20 mg/L for six days was *Bacillus thuringiensis* B1(2015b). This bacterium uses a different biodegradation pathway to those described above, involving the enzymes phenol- and hydroquinone-monooxygenases [68]. Degradation is initiated by phenol- and hydroquinone-monooxygenase, hydroxylating ibuprofen’s aromatic ring and aliphatic chain, forming 2-hydroxyibuprofen, 2-(4-hydroxyphenyl)-propionic acid, 1, 4-hydroquinone, and 2-hydroxyquinol (Figure 2). The degradation process requires additional enzymes. Acyl-CoA synthase is required to produce the intermediate 1,4-hydroquinone, which is then converted to 2-hydroxy-1,4-quinol by hydroquinone monooxygenase. Hydroxyquinol 1,2-dioxygenase then catalyzes the ortho-cleavage of 2-hydroxy-1,4-quinol to form 3-hydroxy-cis, cis-muconic acid (Figure 2). Co-contaminants can enhance ibuprofen biodegradation. For example, the ibuprofen biodegradation efficiency of *Bacillus thuringiensis* B1(2015b) improved when phenol and benzoate were added as co-contaminants [68,88].

Two new ibuprofen-degrading bacteria, *Citrobacter freundii* strain PYI-2 (MT039504) and *C. portucalensis* strain YPI-2 (MN744335), have been isolated. They produce the metabolic intermediates hydroxyibuprofen, 2-(4-hydroxyphenylpropionic acid), 1,4-hydroquinone, and 2-hydroxy-1,4-quinol, suggesting that, like *Bacillus thuringiensis* B1(2015b), they use the degradation pathway mentioned above [89] (Figure 2).

The bacteria with ibuprofen biodegrading potential isolated from activated sludge included Gram-positive bacterium of the genus *Patulibacter*. It degrades ibuprofen under laboratory conditions at concentrations of 125, 46, and 31 µg/L during incubation for 300, 90, and 90 h, respectively. The proteins involved in ibuprofen biodegradation were identified using proteomic analysis. They included acyl-CoA synthetase, a protein with a Rieske (2Fe-2S) iron-sulfur cluster, enoyl-CoA hydratase, and ATP-binding cassette (ABC) transporter proteins. The protein with a Rieske (2Fe-2S) iron-sulfur cluster could oxidize aromatic rings. Enoyl-CoA hydratase catalyzes the double-bond hydroxylation after ring union. ABC transporters participate in the transport of compounds [17]. Their identification suggests that they might be involved in ibuprofen degradation by the *Patulibacter* bacterium.

The strain *Nocardioides* carbamazepine sp. nov. (CBZ_1T = NCAIM B.0.2663 = LMG 32395) removed 70% of ibuprofen after seven weeks in a mineral medium supplemented with glucose (3 g/L) and ibuprofen (1.5 mg/L) [90]. However, its ibuprofen degradation pathway was not reported.

*Rhodococcus cerastii* IEGM 1278 is another bacterium that could biodegrade ibuprofen at 0.1 and 100 mg/L after incubation for 30 and 144 h, respectively, using n-hexadecane as a cometabolite. *R. cerastii* IEGM 1278 oxidized ibuprofen, bioconverting it into hydroxylated and decarboxylated derivatives. However, ibuprofen induced a transition from single- to multi-cellular forms [91].

A novel enzymatic-nonenzymatic coupling degradation mechanism has been proposed based on the ubiquitous marine *Pseudoalteromonas* sp. This new ibuprofen degradation mechanism consists of initiating degradation by treatment with extracellular reactive oxygen species to produce the intermediate 4-ethylresorcinol, which is then degraded by intracellular enzymes. Ibuprofen treated with the agent extracellular hydroxyl and superoxide anion radicals and hydrogen peroxide produced 4-ethylresorcinol. Then, 4-ethylresorcinol was degraded intracellularly until mineralization by 4-hydroxyphenylpyruvate dioxygenase, homogentisate 1,2-dioxygenase, long-chain acyl-CoA synthetase, acetyl-CoA acyltransferase, and enoyl-CoA [92].

Therefore, all ibuprofen biodegradation pathways described to date converge at the point of central bacterial metabolism and are mainly integrated with the TCA cycle. The studies that described ibuprofen biodegradation must be confirmed with additional analyses to confirm that environmental ibuprofen concentrations induce the biochemical pathways observed under laboratory conditions and are similarly metabolized.

The advantage of using ibuprofen-degrading bacteria is that the metabolic pathways for ibuprofen degradation are already known, mainly the transformation products generated, since some of them are more toxic than the original ibuprofen. In this way, the toxicity of the products generated from the degradation can be controlled or studied; in addition to the isolated bacteria, the degradation of the drug can be optimized and taken to an industrial level. Another advantage is that it is an economical and environmentally friendly alternative. The disadvantage is that the current studies have been conducted at the laboratory level and have not yet been applied in the environment, so the degradation efficiency of ibuprofen when applied in the environment could decrease. It is important to mention that the elimination of ibuprofen in wastewater is complicated since it is in a mixture with other contaminants; for this reason, it is imperative to apply different technologies together (biodegradation together with non-biological technologies) to completely eliminate this drug and avoid its penetration into the environment in wastewater treatment plants.

## 6. Conclusions

Ibuprofen, considered an emerging contaminant, is present in different ecosystems due to its high consumption, with ever-increasing environmental accumulation. The lack of emerging technologies to control or remove ibuprofen causes its release into wastewater. Moreover, the inefficient elimination process in water treatment plants causes ibuprofen to be present in effluents and even in domestic water. Experimental studies indicate that ibuprofen has toxic effects on aquatic animals and even on other animals and plants. Further research is required to clarify its effects on terrestrial animals and humans. Furthermore, during its degradation process (chemical or biological), the transformation products produced are more toxic than the original compound, exacerbating the environmental contamination problem. Therefore, efficient degradation processes that are non-toxic or less toxic transformation products are needed. Isolating microorganisms that highly efficiently biodegrade ibuprofen and have greater environmental adaptive capacity are needed. Finally, additional confirmatory studies on the metabolic ibuprofen biodegradation pathways are needed to understand and control it as an emerging contaminant.

## Figures and Tables

**Figure 1 molecules-28-02097-f001:**
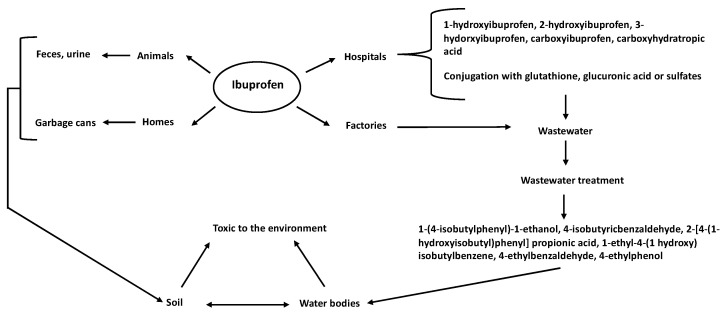
Routes of ibuprofen entry into the environment. Ibuprofen reaches the ground or bodies of water from different sources, such as hospitals, factories, animals and homes.

**Figure 2 molecules-28-02097-f002:**
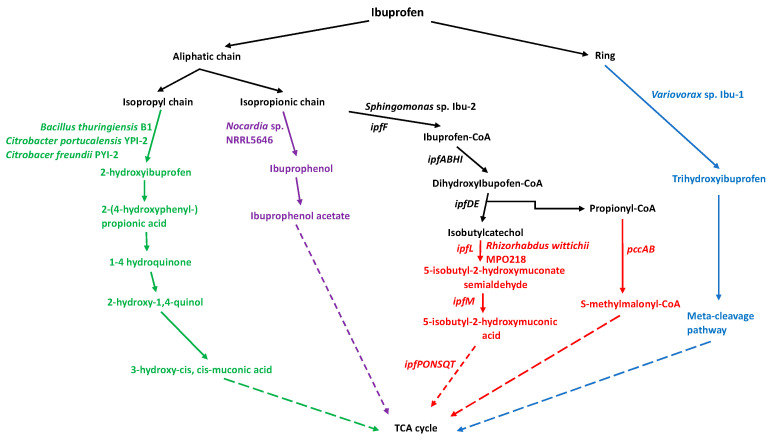
Biodegradation of ibuprofen by bacteria. The colors represent the metabolic pathways used by each bacterial genus. Dotted lines mean that there are more steps during the biodegradation process.

## Data Availability

Not applicable.

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
