# Peer review of "Ibuprofen: Toxicology and Biodegradation of an Emerging Contaminant"

_molecules, 2023, doi:10.3390/molecules28052097_

Round 1

Reviewer 1 Report

The manuscript (molecules-2215585) described the environmental presence, aquatic toxicity, biodegradation pathways, and degrading bacteria of a typical anti-inflammatory drug ibuprofen.

Please supplement toxicity studies of ibuprofen based on the original Title, which were not limited to the aquatic environment, e.g., [1].

[1] Zur, J., Pinski, A., Marchlewicz, A., Hupert-Kocurek, K., Wojcieszynska, D., Guzik, U., 2018. Organic micropollutants paracetamol and ibuprofen-toxicity, biodegradation, and genetic background of their utilization by bacteria, Environ. Sci. Pollut. Res. 25, 21498-21524. http://dx.doi.org/10.1007/s11356-018-2517-x.

More data representation should be added to the Abstract with previous studies to highlight the significance and value of this study. What’s the half-lives of ibuprofen? How to remove ibuprofen more effectively?

The toxicology and biodegradation of ibuprofen should be emphasized from the chirality, which is an essential characteristic for the target compound. Enantiomers of chiral compounds, in spite of their very similar structure and appearance as a single compound in conventional analyses, can show different toxicity, excretion, metabolism, and bioactivities when exposed to asymmetric chemical or biological systems.

   The writing structure and language description should be refined carefully by authors. The published data should be further integrated to specific sections of this manuscript for the comprehensive assay, which is the powerful support for the conclusion, e.g., [2].

[2] Ferrando-Climent, L., Collado, N., Buttiglieri, G., Gros, M., Rodriguez-Roda, I., Rodriguez-Mozaz, S., Barcelo, D., 2012. Comprehensive study of ibuprofen and its metabolites in activated sludge batch experiments and aquatic environment, Sci. Total Environ. 438, 404-413. http://dx.doi.org/10.1016/j.scitotenv.2012.08.073.

Author Response

We appreciate your valuable comments which undoubtedly enriched the manuscript, and we proceed to answer your concerns.

Point 1: Please supplement toxicity studies of ibuprofen based on the original Title, which were not limited to the aquatic environment, e.g., [1].[1] Zur, J., Pinski, A., Marchlewicz, A., Hupert-Kocurek, K., Wojcieszynska, D., Guzik, U., 2018. Organic micropollutants paracetamol and ibuprofen-toxicity, biodegradation, and genetic background of their utilization by bacteria, Environ. Sci. Pollut. Res. 25, 21498-21524. http://dx.doi.org/10.1007/s11356-018-2517-x.

Response 1: Following your recommendation not to limit the study only to aquatic environments, other studies on the toxicity of ibuprofen in other animals and plants were added. Please check section 3, the lines 254-283.

Point 2: More data representation should be added to the Abstract with previous studies to highlight the significance and value of this study. What’s the half-lives of ibuprofen? How to remove ibuprofen more effectively?

Response 2: In the summary of the review, we provide general data on ibuprofen, mainly as an emerging contaminant; the details are shown throughout the manuscript, for this reason we do not consider it appropriate to go deeper with more data in this summary section.

Point 3: The toxicology and biodegradation of ibuprofen should be emphasized from the chirality, which is an essential characteristic for the target compound. Enantiomers of chiral compounds, in spite of their very similar structure and appearance as a single compound in conventional analyses, can show different toxicity, excretion, metabolism, and bioactivities when exposed to asymmetric chemical or biological systems.

Response 3: This observation was addressed and a paragraph on the ibuprofen’ chirality and its relationship to toxicity was added. Please check section 3, lines 295-321.

Point 4: The writing structure and language description should be refined carefully by authors. The published data should be further integrated to specific sections of this manuscript for the comprehensive assay, which is the powerful support for the conclusion, e.g., [2].

[2] Ferrando-Climent, L., Collado, N., Buttiglieri, G., Gros, M., Rodriguez-Roda, I., Rodriguez-Mozaz, S., Barcelo, D., 2012. Comprehensive study of ibuprofen and its metabolites in activated sludge batch experiments and aquatic environment, Sci. Total Environ. 438, 404-413. http://dx.doi.org/10.1016/j.scitotenv.2012.08.073.

Response 4: In this new version of the manuscript, the writing structure and language description was refined. Example [2], due to its importance, was added in different parts of the manuscript. Lines 146-153 and lines 199-203.

Reviewer 2 Report

The anti-inflammatory drug ibuprofen is widely used during the public health event of COVID-19, inevitably, it become a potential environmental contaminant. This review focuses on deepening and updating the information on ibuprofen as a potential emerging environmental contaminant and the potential use of bacteria for its biodegradation as an alternative technology. In addition, the bacterial biodegradation pathway of ibuprofen is discussed. The authors did extensive literature review and in-depth discussion, and it is a meaningful work. This manuscript can be accepted for publication after minor revisions to address the following issues:

1. How to solve the problem that the special conditions of microbial degradation of ibuprofen rarely appear in the environment mentioned in this paper?Can you provide some constructive suggestions?

2. What are the advantages and disadvantages of different ibuprofen degrading bacteria mentioned in the fifth part?

3. It would be better  supplying some discussions about the adaptation conditions of different  ibuprofen removal technologies. For example, some photocatalytic degradation technologies, Applied Surface Science, 583 (2022) 152565; Applied Catalysis B: Environmental, 311 (2022) 121363.

4. The width of the figures should not exceed the width of the paper on the page.

5. In the second paragraph of the introduction, can you give the ibuprofen consumption of Norway and Denmark?

6. Adjustment of some units: The units of "321–610μg/kg to 0.213μg/L" in lines 144 to 145 inconsistent, it is suggested to be unified; In line 223 of this paper, 14200μg/Lcan be changed to 14.2mg/L.

Author Response

We appreciate your valuable comments which undoubtedly enriched the manuscript, and we proceed to answer your concerns.

Point 1: How to solve the problem that the special conditions of microbial degradation of ibuprofen rarely appear in the environment mentioned in this paper?Can you provide some constructive suggestions?

Response 1: In section 4, lines 339-347, some suggestions were added on how to solve the problem of environmental conditions.

Point 2: What are the advantages and disadvantages of different ibuprofen degrading bacteria mentioned in the fifth part?

Response 2: In the lines 514-527, a paragraph about this question was added.

Point 3: It would be better supplying some discussions about the adaptation conditions of different ibuprofen removal technologies. For example, some photocatalytic degradation technologies, Applied Surface Science, 583 (2022) 152565; Applied Catalysis B: Environmental, 311 (2022) 121363.

Response 3:  In lines 157-167, the adaptation conditions of the different ibuprofen removal technologies were added, including photocatalysis and the manuscript that supports it.

Point 4: The width of the figures should not exceed the width of the paper on the page.

Response 4: The figures were adjusted.

Point 5: In the second paragraph of the introduction, can you give the ibuprofen consumption of Norway and Denmark?

Response 5: Ibuprofen consumption was added for both countries, please check lines 54-55.

Point 6: Adjustment of some units: The units of "321–610μg/kg to 0.213μg/L" in lines 144 to 145 inconsistent, it is suggested to be unified; In line 223 of this paper, “14200μg/L”can be changed to 14.2mg/L.

Response 6: The units were set to a single measurement system, please check lines 139-141 and lines 248-249.

Reviewer 3 Report

Pharmaceutical compounds, including ibuprofen, have aroused interest in the scientific community for several years. Increasingly, these compounds are being identified in various environmental matrices. Ibuprofen is one of the most commonly used pharmaceutical compounds by humans, which translates into the penetration of this compound into the environment. It is important to monitor the concentrations of this compound, as well as to focus on ibuprofen transformation products due to their potential adverse impact on the environment and humans. The work is a good overview and will be of interest to readers. However, the reviewer proposes the following changes before publication:

·        Line 42 replace „reduces” with reduce

·        Line 47 replace „has led to it being considered „with has become considered

·        „Line 48 This production is distributed across societies that use it for specific medical purposes. „ Please fix this sentence. The production misrepresents the meaning of this sentence.

·        It is worth writing in section 2 about emerging pollutants. These are compounds that are not covered by the monitoring system and consist of different classes of compounds that are worth mentioning. These compounds have been penetrating the environment for a long time, but only the development of knowledge and technology has made it possible to identify these compounds.

·        Line 98 The caption of figure 1 should be concise, e.g. routes of ibuprofen penetration into the environment. The remaining description should be transferred to the main text, taking into account the literature references cited by the authors.

·        Line 182 replace  in in vivo with in vivo

·        Lines 182-183  in vivo, in vitro. Please put words in italics

·        Line 232 replace ratio with the ratio

·        Section 4 describes the biotransformation/biodegradation of ibuprofen. The word metabolites appear in the text, according to the reviewer, a better term would be transformation products.

Author Response

We appreciate your valuable comments which undoubtedly enriched the manuscript, and we proceed to answer your concerns.

Point 1: Line 42 replace „reduces” with reduce

Response 1: Thanks for the correction, which was included in current line 43.

Point 2: Line 47 replace „has led to it being considered „with has become considered

Response 2: The suggestion was made in line 48.

Point 3: Line 48 This production is distributed across societies that use it for specific medical purposes. „ Please fix this sentence. The production misrepresents the meaning of this sentence.

Response 3: The sentence was modified, and we changed “production” to “drug” in the current line 49.

Point 4: It is worth writing in section 2 about emerging pollutants. These are compounds that are not covered by the monitoring system and consist of different classes of compounds that are worth mentioning. These compounds have been penetrating the environment for a long time, but only the development of knowledge and technology has made it possible to identify these compounds.

Response 4: This point was mentioned in the lines 157-167, the technologies with the conditions and efficiencies of ibuprofen degradation are mentioned, as well as comments on their importance in reducing the penetration of this drug into the environment.

Point 5: Line 98 The caption of figure 1 should be concise, e.g. routes of ibuprofen penetration into the environment. The remaining description should be transferred to the main text, taking into account the literature references cited by the authors.

Response 5: The caption of Figure 1 was shortened.

Point 6: Line 182 replace in in vivo with in vivo

Response 6: The text was changed in the current line 194.

Point 7: Lines 182-183 in vivo, in vitro. Please put words in italics

Response 7: The text was changed in the current lines 194-195.

Point 8: Line 232 replace ratio with the ratio

Response 8: It was done in current line 287.

Point 9: Section 4 describes the biotransformation/biodegradation of ibuprofen. The word metabolites appear in the text, according to the reviewer, a better term would be transformation products.

Response 9: "Metabolites" was changed to "transformation products" in section 4 and throughout the manuscript where it was appropriate.

Round 2

Reviewer 1 Report

The manuscript has been revised accordingly, and the responses to the comments and suggestion of reviewers were reasonable.